# Improving microbial fitness in the mammalian gut by *in vivo* temporal functional metagenomics

Stephanie J Yaung[1,2,3], Luxue Deng[4], Ning Li[4], Jonathan L Braff[3], George M Church[2,3], Lynn Bry[4], Harris H Wang[5,6,†,*] & Georg K Gerber[4,†,**]

## Abstract

Elucidating functions of commensal microbial genes in the mammalian gut is challenging because many commensals are recalcitrant to laboratory cultivation and genetic manipulation. We present Temporal FUnctional Metagenomics sequencing (TFUMseq), a platform to functionally mine bacterial genomes for genes that contribute to fitness of commensal bacteria *in vivo*. Our approach uses metagenomic DNA to construct large-scale heterologous expression libraries that are tracked over time *in vivo* by deep sequencing and computational methods. To demonstrate our approach, we built a TFUMseq plasmid library using the gut commensal *Bacteroides thetaiotaomicron* (Bt) and introduced *Escherichia coli* carrying this library into germfree mice. Population dynamics of library clones revealed Bt genes conferring significant fitness advantages in *E. coli* over time, including carbohydrate utilization genes, with a Bt galactokinase central to early colonization, and subsequent dominance by a Bt glycoside hydrolase enabling sucrose metabolism coupled with co-evolution of the plasmid library and *E. coli* genome driving increased galactose utilization. Our findings highlight the utility of functional metagenomics for engineering commensal bacteria with improved properties, including expanded colonization capabilities *in vivo*.

**Keywords** commensal fitness; functional metagenomics; microbiota; next-generation sequencing; synthetic biology
**Subject Categories** Synthetic Biology & Biotechnology; Chromatin, Epigenetics, Genomics & Functional Genomics
**Mol Syst Biol. (2015) 11: 788**

See also: **JJ Faith** *et al* (March 2015)

## Introduction

The mammalian gastrointestinal (GI) tract is a hostile environment for poorly adapted microbes. Nonetheless, diverse groups of microbes have evolved to prosper in the GI tract, in the setting of intense interspecies competition, physical and chemical stressors, and the host immune system (Ley *et al*, 2006; Dethlefsen *et al*, 2007). These microorganisms also support the normal homeostatic functions of the host by helping to extract nutrients, stimulate the immune system, and provide protection against colonization by pathogens (Stappenbeck *et al*, 2002; Hooper, 2004; Bäckhed *et al*, 2005; Gill *et al*, 2006; Ley *et al*, 2006). Next-generation sequencing has enabled systematic studies of the mammalian microbiota, and great strides have been made in characterizing the structure of bacterial communities and their genetic potential *in vivo*. For instance, the Human Microbiome Project (HMP) (Turnbaugh *et al*, 2007; Peterson *et al*, 2009; Huttenhower *et al*, 2012) and MetaHIT (Qin *et al*, 2010) have generated maps of bacterial species abundances throughout the human body, reference genomes, and catalogs of more than 100 million microbial genes assembled from shotgun sequencing of *in vivo* communities. Although these studies have generated vast amounts of descriptive data, the functions of most bacterial genes in these collections remain poorly characterized or wholly unknown.

Traditional methods to characterize the functions of microbial genes require the isolation, cultivation, and introduction of foreign DNA into a recipient organism. However, an estimated 60–80% of mammalian-associated microbiota species remain uncultivated (Walker *et al*, 2014). Even after successful culture and introduction of genetic material into a microorganism, the DNA must integrate into the microbial genome or be maintained episomally. This requires known compatible replication and restriction–modification systems, which may not be feasible for many microbes. If these barriers can be overcome, standard low-throughput methods for functional characterization of genes may be employed, or newer

1 Program in Medical Engineering and Medical Physics, Harvard-MIT Division of Health Sciences and Technology, Massachusetts Institute of Technology, Cambridge, MA, USA
2 Department of Genetics, Harvard Medical School, Boston, MA, USA
3 Wyss Institute for Biologically Inspired Engineering, Harvard University, Boston, MA, USA
4 Center for Clinical and Translational Metagenomics, Department of Pathology, Brigham & Women's Hospital, Harvard Medical School, Boston, MA, USA
5 Department of Systems Biology, Columbia Initiative in Systems Biology, Columbia University, New York, NY, USA
6 Department of Pathology and Cell Biology, Columbia University Medical Center, New York, NY, USA
*Corresponding author. Tel: +1 212 305 1697; E-mail: hw2429@columbia.edu
**Corresponding author. Tel: +1 617 278 0468; E-mail: ggerber@partners.org
†These authors contributed equally to this work

approaches such as transposon mutagenesis could be coupled with next-generation sequencing. In this latter method, random locations on the genome are disrupted with a transposon containing a selectable marker; the resulting library is subjected to selection conditions and deep sequenced to determine enriched and depleted mutants (Van Opijnen *et al*, 2009). A limitation of this technique is that essential genes or those that are important to cell fitness are difficult to assay, since inactivation of these genes by transposon mutagenesis would be lethal to the organism under study. An additional constraint is that transposon mutagenesis may disrupt the expression of bystander genes that are near the relevant locus, thus causing confounding phenotypic effects.

Here, we employ an alternative approach, by building large-scale shotgun expression libraries that can confer a gain of function in the recipient bacterial strain. Our method uses physical shearing or restriction digestion of donor DNA to generate fragments that are cloned into an expression vector and transformed into the recipient bacterial strain, for high-throughput functional screening to identify genes that confer a fitness advantage in a particular context. This approach has the advantage that the donor organism need not be readily culturable or genetically manipulable in the laboratory; moreover, it allows the investigation of essential genes or those conferring a fitness advantage synergistic with the recipient organism. Functional metagenomics using environmental samples was first established for communities derived from lignocellulosic feedstocks (Healy *et al*, 1995), seawater (Stein *et al*, 1996), and soil (Rondon *et al*, 2000). The use of shotgun libraries for functional metagenomics of mammalian-associated microbiota has been demonstrated *ex vivo*, such as by growing the library in media with different substrates to characterize carbohydrate active enzymes (Tasse *et al*, 2010), prebiotic metabolism (Cecchini *et al*, 2013), glucuronidase activity (Gloux *et al*, 2011), salt tolerance (Culligan *et al*, 2012), and antibiotic resistance genes (Sommer *et al*, 2009), or by using filtered lysates of the library to screen for signal modulation in mammalian cell cultures (Lakhdari *et al*, 2010). This metagenomic shotgun library approach has yet to be carried out on a large scale *in vivo*.

To demonstrate our Temporal FUnctional Metagenomics sequencing (TFUMseq) approach, we used high-coverage genetic fragments from the genome of the fully sequenced human gut commensal *Bacteroides thetaiotaomicron* (Bt) (Xu *et al*, 2003) and cloned the fragments into a plasmid library in an *Escherichia coli* K-12 strain. We chose Bt because it is a common commensal strain in the human gut that persistently colonizes and possesses a broad and well-characterized repertoire of catabolic activities, such as sensing polysaccharides and redirecting metabolism to forage on host versus dietary glycans (Sonnenburg *et al*, 2005; Bjursell *et al*, 2006; Martens *et al*, 2008). We subjected the TFUMseq library to *in vitro* and *in vivo* selective pressures, collected output samples at different time points for high-throughput sequencing, and used computational methods to reconstruct the population dynamics of clones harboring donor genes (Fig 1). Our work is an advance over previous studies in two major aspects. First, to our knowledge, our study is the first to employ shotgun expression libraries for functional metagenomics *in vivo*. Important features of the mammalian gut are difficult to recapitulate *in vitro*, such as the host immune response. Thus, *in vivo* experiments are essential for investigating the function of commensal microbiota genes in the host. Second, our study

leverages high-throughput sequencing and computational methods to generate detailed dynamics of the entire population subject to selection over time. This kinetic information is crucial for understanding succession events during the inherently dynamic and complex process of host colonization.

## Results

### Library construction and characterization

A 2.2 kb *E. coli* expression vector, GMV1c, was constructed to include the strong constitutive promoter pL and a ribosomal binding site upstream of the cloning site for input DNA fragments (Fig 1). We cloned in 2–5 kb fragments of donor genomic DNA from Bt and generated a library of ~100,000 members, corresponding to > 50× coverage of the donor genome. We sequenced the library on the Illumina HiSeq 2500 instrument to confirm sufficient coverage of the Bt genome (Fig 2A and Supplementary Fig S1). The distribution of member insert sizes in the input library was verified to be centered around 2–3 kb (Fig 2B), a size range allowing for the full-length representation of almost all Bt genes.

### *In vitro* stability and selection by media condition

To determine vector stability *in vitro*, we performed serial batch passaging of cells carrying GMV1c every 1–2 days over 2 weeks in two media conditions: aerobic Luria broth (LB) and anaerobic mouse chow filtrate (MC). We expected the MC medium and anaerobic conditions to better reflect aspects of the nutritional content and oxygenation status in the mouse gut than the rich LB medium in aerobic conditions. In both conditions, the vector was maintained in over 80% of library members without antibiotic selection throughout 2 weeks of *in vitro* passaging (~70 generations) (Fig 2C), suggesting general stability of the medium copy vector (~40 copies per cell). Clones harboring the empty vector (i.e. plasmid with no Bt insert) were the most fit library member: In both LB and MC conditions, these clones initially constituted 70% of the library and increased to 90% by the end of 2 weeks, albeit at a slower rate in anaerobic MC (Supplementary Fig S2A).

To identify Bt genes with differential *in vitro* selection in LB and MC conditions relative to the input library, we isolated DNA from Day 0 and Day 6 or 7 cultures, amplified the inserts by PCR for deep sequencing on the Illumina MiSeq platform, and used computational methods to determine donor genes that were differentially enriched or depleted. In each condition, we found a number of significantly enriched Bt genes (Supplementary Table S1). At Day 7 in aerobic LB, enriched genes included metabolic enzymes, such as chitobiase (BT_0865), which degrades chitin, and stress response proteins, such as glycine betaine/L-proline transport system permease (BT_1750), which is involved in the import of osmoprotectants glycine betaine or proline that mitigate effects of high osmolarity (Haardt *et al*, 1995). At Day 6 in anaerobic MC, a different set of genes was significantly enriched, particularly the locus consisting of endo-1,4-beta-xylanase (BT_0369), galactokinase (BT_0370), glucose/galactose transporter (BT_0371), and aldose 1-epimerase (BT_0372). These results highlight that our functional metagenomics approach is able to enrich for likely bioactive donor genes that

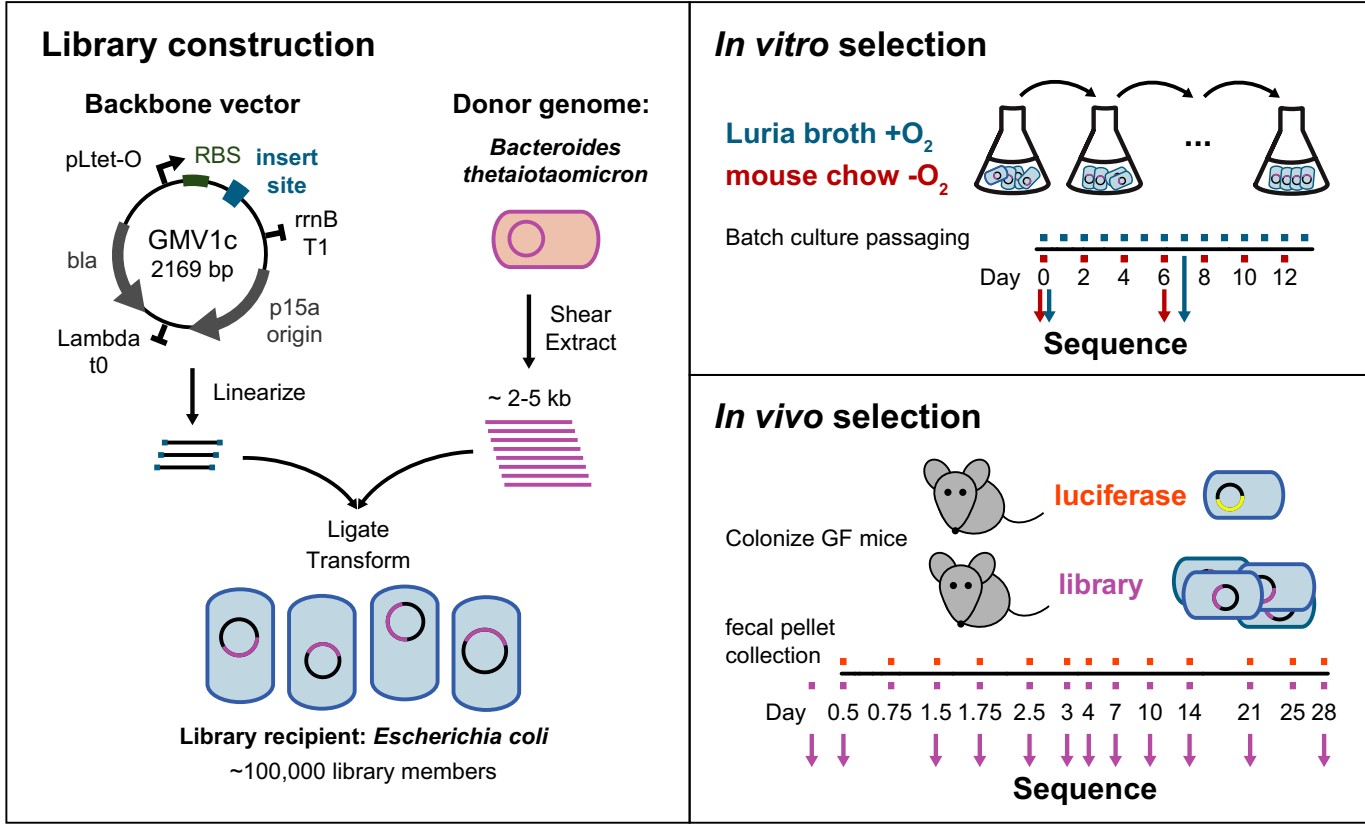

**Figure 1.   Experimental design.**

(Left panel) Map of the library backbone vector. The vector was linearized and ligated to sheared fragments of donor genome to generate the heterologous insert library. (Right panels) Passaging of the *Escherichia coli* library in two liquid media conditions (top) and inoculation of the library or a control luciferase plasmid into germfree (GF) mice (bottom). Small boxes across the time line denote sample collection points. Arrows indicate deep-sequenced samples.

improve fitness of the recipient cells in *in vitro* passaging conditions. Enolase (BT_4572), the only common hit among annotated genes in both media conditions, was found to be depleted relative to the input library. This enzyme catalyzes the penultimate step of glycolysis, and its overexpression may be toxic in *E. coli* (Usui *et al*, 2012).

### *In vivo* library selection in germfree mice

To investigate *in vivo* gene selection in our library, we inoculated two cohorts of C57BL/6 male 6- to 8-week-old germfree mice (*n* = 5 per group) and maintained the mice for 28 days under gnotobiotic conditions. One cohort was colonized with our library and the other cohort with a control GMV1c vector carrying the 5.9-kb luciferase operon (luxCDABE from *Photorhabdus luminescens,* Winson *et al*, 1998). Fecal pellets were collected on days 0.5, 0.75, 1.5, 1.75, 2.5, 3, 4, 7, 10, 14, 21, 25, and 28 after inoculation.

To determine *in vivo* vector stability, we plated fecal pellets on LB, on which *E. coli* either with or without vectors would grow, and on LB + carbenicillin, selective for *E. coli* harboring our vectors. Strains carrying the luciferase vector dropped by ~100,000-fold by Day 28 compared to the earliest plated time point (18 h), presumably due to negative selective pressures from the energy consumption of the vector-borne luciferase in *E. coli* (Fig 3A). In contrast, our library was well maintained *in vivo* throughout the 28 days of

the experiment, suggesting at least minimal fitness cost to maintain the Bt insert library. Furthermore, unlike in the *in vitro* experiment, where clones containing the empty vector were enriched over time, these clones were virtually absent by the end of the *in vivo* experiments (Supplementary Fig S2B), suggesting positive selection had taken place.

### Characterization of *in vivo* library population dynamics

To characterize the entire *in vivo* selected library over time, we extracted DNA from collected stool samples, PCR-amplified the donor inserts, prepared sequencing libraries of the amplicons, sequenced libraries on the Illumina HiSeq 2500 instrument, and used computational techniques to detect selected genes in the donor genome that were uniformly covered over time by more than the expected background number of sequencing reads. Each sample resulted in ~7 million 101 nt paired-end reads (Supplementary Table S4) that were mapped back to the donor genome (Supplementary Fig S6). We also employed Sanger sequencing of vectors from clones directly isolated from stool samples to confirm deep-sequencing results and obtain insights into the structure of full-length inserts.

To obtain a genome-wide view of library selection over time and across the different mice, we calculated an information theoretic

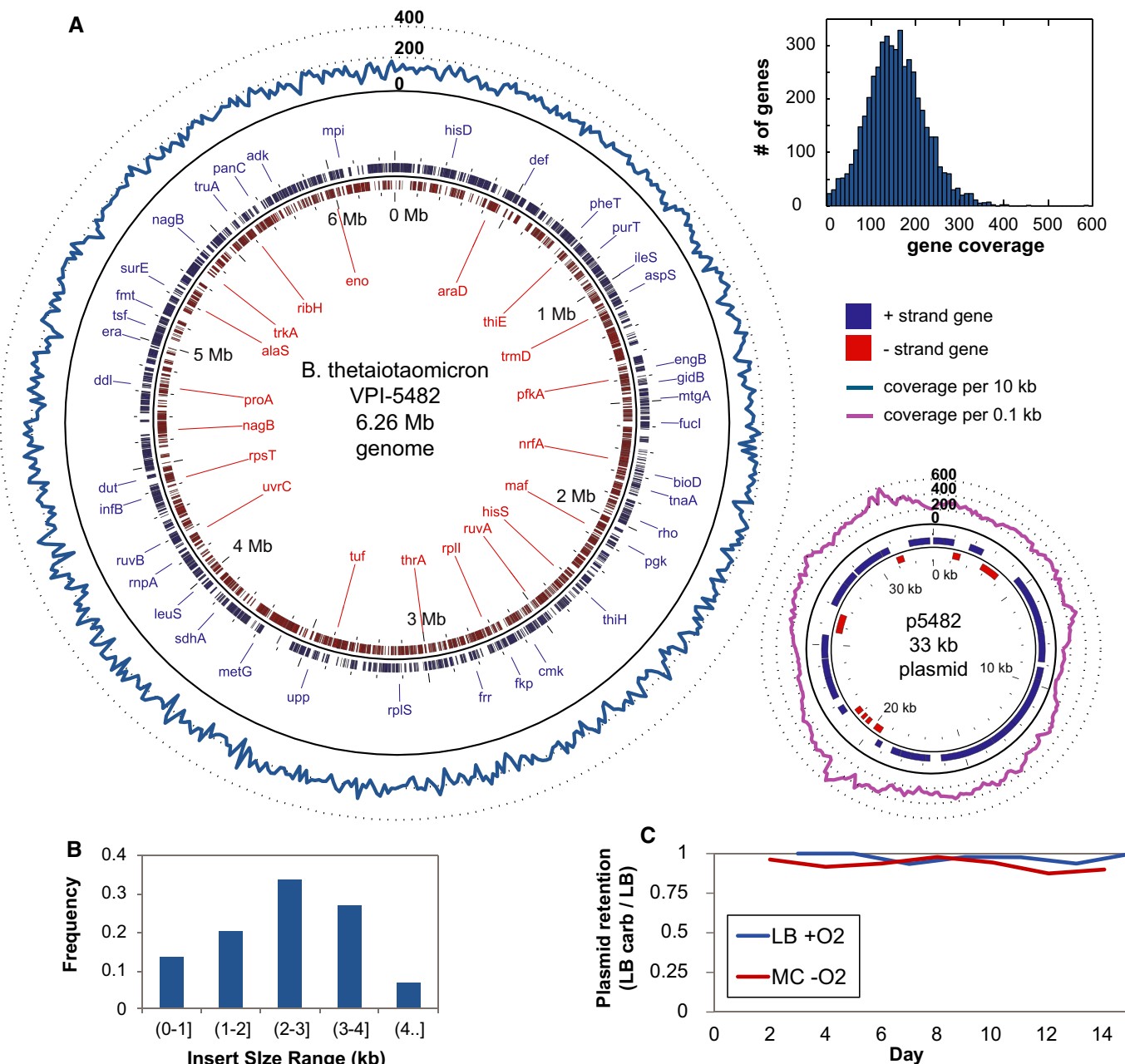

**Figure 2. Input library characterization.**

A   Even coverage of the *Bacteroides thetaiotaomicron* genome. The blue and purple lines represent per-base coverage values for the chromosome and native *B. thetaiotaomicron* p5482 plasmid, respectively. The histogram (top right) shows the distribution of genes by their coverage (normalized to gene length).

B   Insert size distribution of library.

C   Plasmid retention calculated by comparing number of colonies on LB versus LB + carbenicillin plates from *in vitro* passaging experiments in aerobic LB or anaerobic mouse chow (MC) filtrate.

Source data are available online for this figure.

measure, termed effective positional diversity, similar to that commonly used to quantify population diversity in macroscopic and microscopic ecology studies (Jost, 2006; Schloss *et al*, 2009) (Fig 3B). This measure, equal to the exponentiated Shannon entropy over all positions in the Bt genome, reflects how many positions in the donor genome are evenly represented in the population.

Effective positional diversity values of the initial library were ~6 Mb, indicating essentially even coverage of the entire Bt genome. From Day 1.75 to Day 7 and continuing until the end of the experiment at Day 28, there was a rapid decline in effective positional diversity, which signifies expansion in the population of clones harboring inserts at a limited number of Bt genomic loci.

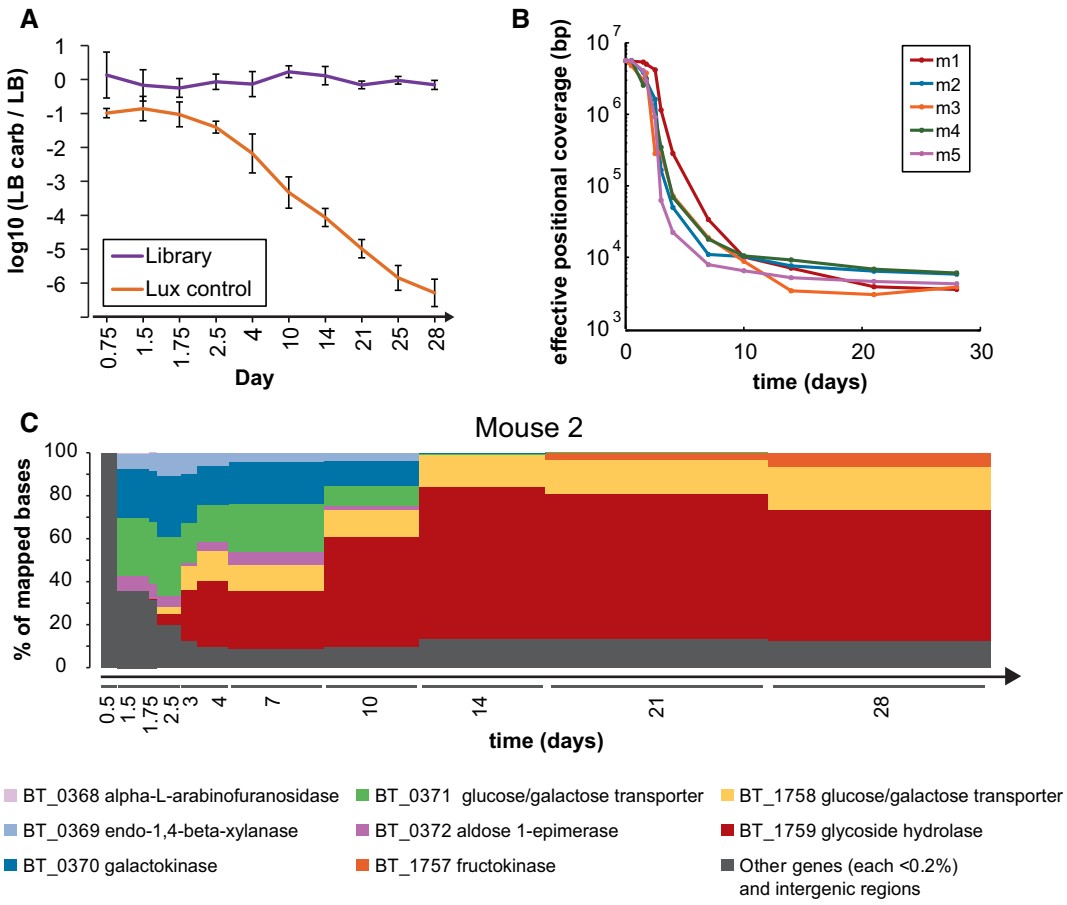

**Figure 3.** ***In vivo*** **selection experiments.**

A  Plasmid retention calculated by comparing number of colonies on LB versus LB + carbenicillin plates from mouse fecal samples. *n* = 5 mice; error bars = standard deviation.

B  Effective positional coverage across the entire Bt genome for each mouse (m1–m5) begins with essentially even coverage of the Bt genome of ~6 Mb, but drops rapidly over the experimental time-course, representative of selection at specific loci.

C  Representative longitudinal selection of Bt genes in a single mouse (Mouse 2). For each mouse and time point, ~$10^9$ sequenced bases were mapped to the *B. thetaiotaomicron* genome. Of those mapped bases, the percentage mapping to each gene is plotted. Genes with < 0.2% are grouped together (dark gray bars).

Source data are available online for this figure.

To explore the kinetics of gene selection *in vivo*, we plotted the percentage of sequencing reads mapped to genes in the *Bt* genome over time and examined genes constituting > 0.2% of total reads. As noted, prior to inoculation, the read coverage was even over the entire Bt genome and corresponded to < 0.2% per gene. Figure 3C provides a representative visualization of Bt gene selection for Mouse 2 (plots for other mice are shown in Supplementary Fig S3). By 36 h post-inoculation, five genes, alpha-L-arabinofuranosidase, endo-1,4-beta-xylanase, galactokinase, glucose/galactose transporter, and aldose 1-epimerase (BT_0368 to BT_0372), comprised over half of the reads mapped. At Day 2.5, glucose/galactose transporter (BT_1758) and glycoside hydrolase (BT_1759) became noticeable and continued to increase until they saturated all reads at Day 14. Then, fructokinase (BT_1757) emerged and stabilized at around 6% of the reads throughout the remaining 2 weeks of the experiment. These observations are generally consistent across all five mice, though the selection kinetics varied slightly (Supplementary Fig S3). For example, the transition from galactokinase and glucose/galactose transporter (BT_0370 and BT_0371) to glycoside hydrolase (BT_1759) occurred 4 days earlier in Mouse 5 than in Mouse 2, and the emergence of fructokinase (BT_1757) was detectable only in Mice 2, 4, and 5.

In terms of functional groups rather than individual genes, of the 51.4% Bt genes with COG annotations, those related to carbohydrate transport and metabolism comprised 10% of the input library. Averaged across the five mice, these carbohydrate transport and metabolism genes increased to 25% of reads on Day 0.5, 72% on Day 1.5, and essentially 100% by Day 7 (Supplementary Fig S4), suggesting the importance of carbohydrate transport and metabolism in *in vivo* fitness.

To rigorously determine the Bt genes that were differentially represented in the population over time and to localize putatively selected regions to specific genes, we applied information theoretic and statistical techniques for longitudinal analysis (Bar-Joseph *et al*, 2003). In our analyses, transient dominance of clones *in vivo* is of particular interest as different genes may

confer fitness advantages at distinct stages of host colonization. Further, our experiments capture competition among ~100,000 strains harboring distinct genetic fragments, rather than traditional binary competition experiments. Thus, we are interested in not only clones harboring Bt fragments that show an increase over time in relative abundance, but also those clones that show a significantly slower rate of depletion than other clones. To methodically detect these effects, for every Bt gene, we computed two measures: (i) time-averaged relative abundance (TA-RA) and (ii) time-averaged normalized effective coverage (TA-NEC). The TA-RA value is conceptually similar to a time-integrated pharmacological dose value (Byers & Sarver, 2009); in our analysis, it represents the average "dose" of a particular donor gene, relative to all other donor genes present *in vivo* over a period of time. The TA-NEC value quantifies the fraction of the gene that is effectively covered by reads over a period of time. These measures are important to evaluate in tandem, since bystander genetic loci may be differentially abundant in clones (i.e. high TA-RA values) simply because they are contiguous with genes under selection; however, these loci are likely to be detectable as spurious (i.e. low TA-NEC values) because they will often include only fragments of genes.

## Genes showing transient selection during early gut colonization

We found 13 Bt genes during the early stage of gut colonization (up to Day 4) with significantly larger than expected TA-RA and TA-NEC values (*q*-values < 0.05; Table 1). These genes include those coding for enzymes involved in the synthesis of extracellular capsular polysaccharides and lipopolysaccharides (LPS), specifically D-glycero-alpha-D-manno-heptose-1,7-bisphosphate 7-phosphatase (*gmhB*) (BT_0477) and dTDP-4-dehydrorhamnose reductase (*rfbD*; *rmlD*) (BT_1730). There are two biosynthesis pathways of nucleotide-activated *glycero-manno*-heptose that result in either L-β-D-heptose or D-α-D-heptose, which serve as precursors or subunits in LPS, S-layer glycoproteins, and capsular polysaccharides (Valvano *et al*, 2002). The *E. coli* GmhB is critical for complete synthesis of the LPS core (Kneidinger *et al*, 2002). The selection for Bt *gmhB* could allow *E. coli* to expand its extracellular glycoprotein display, since *E. coli* GmhB is highly selective for β-anomers while Bt GmhB prefers α-anomers during hydrolysis of D-*glycero*-D-*manno*-heptose 1β,7-bisphosphate (Wang *et al*, 2010). BT_1730 (*rfbD*; *rmlD*) is involved in dTDP-rhamnose biosynthesis involved in the production of O-antigen, a repetitive glycan polymer in LPS, and potentially other cell-membrane components. Deletion of *rmlD* in *Vibrio cholera* results in a severe defect in the colonization of an infant mouse model (Chiang & Mekalanos, 1999), and uropathogenic *E. coli* lacking functional RmlD lose serum resistance (Burns & Hull, 1998). Thus, expressing Bt *rmlD* could allow the recipient *E. coli* to alter its antigenicity or resistance to host factors that would impede its initial colonization of the mammalian gut.

Several other genes with membrane-associated functions also showed increased selection at Day 4, including outermembrane lipoprotein SilC (BT_0297), cell surface protein (BT_1771), and outermembrane protein OmpA (BT_1511). These genes could confer increased capabilities for *E. coli* to attach to the mucosal surface of the mammalian GI tract or increased adaptations to the gut chemical

**Table 1.    Statistical testing of *in vivo* selection of Bt genes.**

| Gene | Annotation | TA-RA *q*-value | TA-NEC *q*-value |
|------|-----------|-----------------|-------------------|
| BT_0297 | outer membrane lipoprotein SilC | 3.06E-02 | 4.08E-04 |
| **BT_0370** | galactokinase | 1.14E-03 (3.25E-03) | 5.94E-06 (6.95E-09) |
| **BT_0371** | glucose/galactose transporter | 1.14E-03 (3.14E-03) | 3.50E-02 (4.21E-05) |
| BT_0477 | D-glycero-alpha-D-manno-heptose-1,7-bisphosphate 7-phosphatase (*gmhB*) | 1.67E-02 | 1.32E-02 |
| BT_0478 | hypothetical protein | 1.77E-03 | 2.47E-02 |
| BT_1510 | hypothetical protein | 3.86E-02 | 1.10E-03 |
| BT_1511 | outermembrane protein OmpA | 4.38E-02 | 7.33E-04 |
| BT_1730 | dTDP-4-dehydrorhamnose reductase (*rfbD*; *rmlD*) | 3.86E-02 | 3.45E-04 |
| BT_1731 | hypothetical protein | 4.38E-02 | 8.80E-03 |
| BT_1757 | fructokinase | 1.58E-02 | 2.50E-03 |
| **BT_1759** | glycoside hydrolase | 1.19E-02 (2.48E-07) | 2.58E-04 (1.21E-09) |
| BT_1771 | cell surface protein | 4.38E-02 | 4.00E-03 |
| BT_4265 | GMP synthase (*guaA*) | 3.86E-02 | 3.33E-02 |

Genes demonstrating significant *in vivo* selection profiles were determined via statistical testing of time-averaged relative abundance (TA-RA) and time-averaged normalized effective coverage (TA-NEC) values up to either Day 4 or Day 28 of host colonization. Genes showing significant selection up to Day 4 are in white. Genes showing significant selection up to both Day 4 and Day 28 are highlighted in bold. *q*-values for Day 28 are listed in parentheses.

environment. For instance, *Bacteroides fragilis* lacking OmpA are more sensitive to SDS, high salt, and oxygen exposure (Wexler *et al*, 2009). In *Bacteroides vulgatus*, OmpA additionally plays a role in intestinal adherence (Sato *et al*, 2010) and, in *Klebsiella pneumoniae*, activates macrophages (Soulas *et al*, 2000).

Since nucleotide pools are tightly controlled in *E. coli* (Mehra & Drabble, 1981), the selection for Bt GMP synthase *guaA* (BT_4265) may substantially affect intracellular guanine concentration, translation regulation, and cell signaling. Inhibiting GMP synthase induces stationary-phase genes in *Bacillus subtilis* (Ratnayake-Lecamwasam *et al*, 2001), and nucleotide concentrations drop when *E. coli* transition from growth to stationary phase (Buckstein *et al*, 2008). These observations suggest that a copy of heterologous *guaA* could enable escape of native tight regulation of the guanine pool to prolong the cell's exponential growth phase. Moreover, extra GMP synthase may further protect *E. coli* from incorporating mutagenic deaminated nucleobases that would interfere with RNA function and gene expression (Pang *et al*, 2012).

## Genes showing long-term selection during gut colonization

We found three Bt genes over the entire period of colonization (up to Day 28) with significantly larger than expected TA-RA and TA-NEC values (*q*-values < 0.05; Table 1); these genes also showed significant selection during early colonization (up to Day 4). All

three genes are involved in sugar metabolism and transport, suggesting they may act to unlock more nutrient resources for *E. coli* in the gut. We performed *in vitro* experiments, described below, to further characterize the functions of these strongly selected loci, centered around a Bt glycoside hydrolase (BT_1759) and galactokinase (BT_0370).

### Glycoside hydrolase (BT_1759)

From Days 1.5 to 3 in the high-throughput sequencing data, we observed sharply positive selection of glycoside hydrolase (BT_1759), which stabilized and continued to be strongly selected for from Days 4 to 28 across all mice (Fig 4A). We confirmed these results with Sanger sequencing, which additionally allowed us to identify exact junctions and directionality of isolated inserts. In clones from Days 7, 14, and 28, we observed the primary selected insert to be 2.5 kb in length, beginning four nucleotides after the annotated glycoside hydrolase (BT_1759) start codon and ending about one-third of the way into the downstream gene (glucose/galactose transporter). Notably, we also detected other inserts containing different 5′ truncated versions of the glycoside hydrolase in the late time points, both in our high-throughput and Sanger sequencing data (Fig 4B).

Sonnenburg *et al* (2010) previously demonstrated that periplasmic BT_1759 in Bt hydrolyzes smaller fructooligosaccharides and sucrose. To functionally characterize BT_1759 and surrounding genes when heterologously expressed in *E. coli*, we cloned the CDS of each into the backbone vector and transformed it into the starting *E. coli* strain. None of the full-length genes conferred growth in M9 minimal media with sucrose as the sole carbon source (Fig 4C). However, clones isolated from mice on Day 28 were able to metabolize sucrose. Furthermore, retransformation of the DNA vectors from these clones into the starting *E. coli* strain also conferred growth on sucrose, indicating that the phenotype was plasmid-borne. Interestingly, sucrose utilization was enabled when we reconstituted the 4 nt truncation found in many of the Days 7 and 28 Sanger-sequenced clones into the starting *E. coli* strain. These results suggest that the truncation allows for appropriate processing of the signal sequence to express and localize the Bt enzyme in the periplasmic space of *E. coli*, where sucrose is capable of entering by diffusion.

### Galactokinase (BT_0370), glucose/galactose transporter (BT_0371), and native galactokinase reversion

In contrast to the selection profile of glycoside hydrolase (BT_1759), the galactokinase (BT_0370) and glucose/galactose transporter (BT_0371) exhibited an earlier increase in abundance that peaked at Day 2.5 and gradually declined over the remainder of the experiment (Fig 5A). We observed a similar trend in Day 7 clones by Sanger sequencing, with no clones containing BT_0370 or BT_0371 present at Day 28 (Fig 5B).

We confirmed that individually cloned BT_0370 and BT_0371 genes confer galactose utilization in the starting *E. coli* strain when grown using M9 minimal media supplemented with 0.5% galactose as the sole carbon source (Fig 5C). To our surprise, *E. coli* isolated from mouse stool at later time points were able to grow on galactose even though they carried plasmids with glycoside hydrolase (BT_1759), and not the Bt galactose utilization genes (BT_0370 and BT_0371). However, strains retransformed with BT_1759 were

unable to grow on galactose, suggesting that the stool-isolated strains gained the capability to use galactose through mutations independent of the expression plasmid, namely in the recipient *E. coli* genome. After confirmation that our starting *E. coli* strain was galK− due to the presence of an insertion sequence (IS2), we hypothesized that stool isolates reverted to galK$^+$ via loss of IS2. In stool-isolated clones from Days 7, 14, and 28, we found that the galK reversion occurred after Day 7 and was found in > 75% of clones in four of five mice at Day 14 (Fig 5D). Interestingly, *E. coli* harboring the insert library exhibited accelerated galK reversion in the mouse gut; in the luciferase control mice, there was an overall reversion rate of only 50% by Day 28, as opposed to 100% in the mice that had been inoculated with the Bt library. The genomic galK reversion by ~Day 14 suggests that there is early selection for Bt galactokinase (BT_0370), but this foreign gene is subsequently lost as the recipient *E. coli* regain native galactokinase activity, which seems to have a fitness advantage over the heterologously expressed Bt galactokinase gene.

### *In vivo* genomic stability of *E. coli* recipient strain

Given the observed genomic galK reversion, we investigated whether other changes occurred in the *E. coli* genome over the course of our *in vivo* experiments. Genomic stability of bacterial cells in the gastrointestinal tract *in vivo* has not been characterized in great detail, and microbial mutation rates *in vivo* are also not well characterized. We performed whole-genome sequencing of 13 *E. coli* isolates from stool of mice inoculated with the library and two *E. coli* isolates from stool of mice inoculated with the control luciferase construct. Of the isolates from mice that were inoculated with the library, seven were from Day 7 samples containing either BT_1759 or BT_0370 inserts and six were from Day 28 samples containing BT_1759 inserts. In addition to searching for variants in the *E. coli* genome, we also looked for variants on the library plasmid with the known insert locus, and the F plasmid, which was present in the starting *E. coli* strain. Overall, we found single-nucleotide variants (SNVs) in only three of the 15 isolates (Table 2).

One of the isolates from a luciferase control mouse harbored three mutations. One SNV was in the coding sequence of adenylate cyclase *cyaA*, while the other two SNVs were in intergenic regions, between tRNAs *lysW* and *valZ*, and between *traJ* and *traY* on the F plasmid. The functional effects of these SNVs, if any, are unclear. The operon structure of the tRNA region may be *lysT-valT-lysW-valZ-lysYZQ* (Blattner *et al*, 1997), or *valZ-lysY* could be a separate operon as predicted in EcoCyc (Keseler *et al*, 2011), in which case the SNV could affect transcription of the downstream tRNAs. As for the *traY* promoter variant, the -35 hexamer has been documented to be TTTACC (Gaudin & Silverman, 1993). The SNV T > C changes it to CTTACC, which could weaken the promoter to decrease the expression of TraY, a DNA-binding protein involved in the initiation of DNA transfer during conjugation.

One *E. coli* isolate from a library-inoculated mouse had a genomic change that conferred increased growth on galactose. Isolate 1 from Mouse 1 on Day 28 had a mutation in the lactose/melibiose:H$^+$ symporter, *lacY* (F27S), which is a missense mutation in the first transmembrane region (Guan *et al*, 2002). We did not observe phenotypic differences on MacConkey lactose plates, since the *E. coli* recipient strain has a deletion in *lacZ*, and thus,

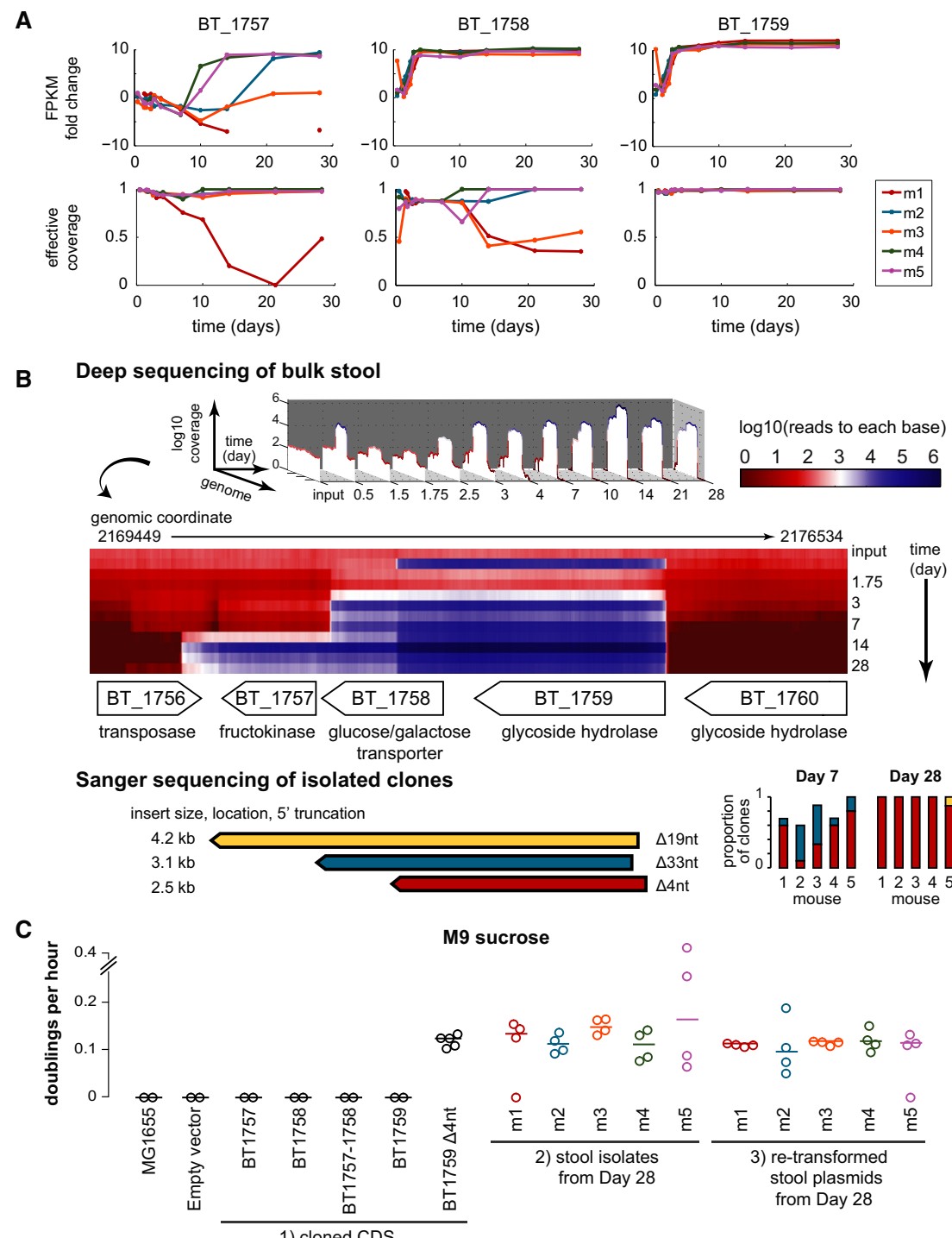

**Figure 4. BT_1759 glycoside hydrolase.**

A   Selection kinetics by fragments per kilobase mapped (FPKM) fold change and normalized effective coverage of genes BT_1757, BT_1758, and BT_1759. m1–5 = Mouse 1–5.

B   Mapped reads to each base in the region with deep sequencing and Sanger sequencing of isolated clones (below, length of inserts is to scale to the gene map). Read values are the mean across five mice and were normalized to 1 billion mapped bases per run to compare across time points. Sanger sequencing was performed on ten clones per mouse at Day 7 and eight clones per mouse at Day 28. nt = nucleotides.

C   Functional characterization in minimal media with sucrose as the sole carbon source. Three sets of strains were studied: (i) starting *E. coli* strains transformed with the CDS of each gene cloned into the backbone vector, (ii) *E. coli* clones directly isolated from stool samples, and (iii) starting *E. coli* strains re-transformed with individual plasmids isolated from stool samples. All clones isolated at Day 28 carried the BT_1759 locus. Lines represent the mean.

Source data are available online for this figure.

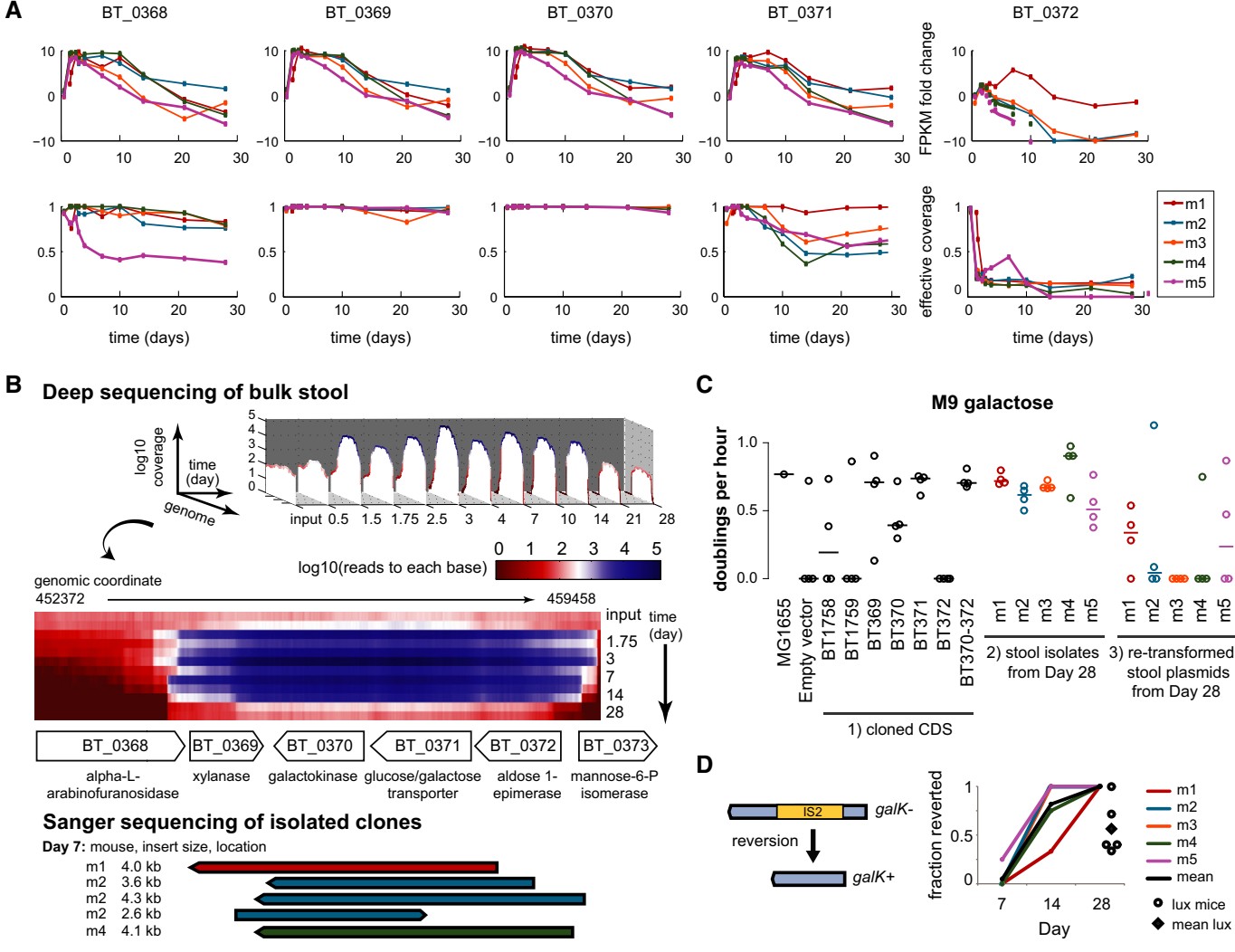

**Figure 5.   BT_0370 galactokinase and BT_371 glucose/galactose transporter.**

A   Selection kinetics by fragments per kilobase mapped (FPKM) fold change and normalized effective coverage of genes BT_0368, BT_0369, BT_0370, BT_0371, and BT_0372.

B   Mapped reads to each base in the region with deep sequencing and Sanger sequencing of isolated clones (below). Read values are the mean across five mice and were normalized to 1 billion mapped bases per run to compare across time points. Isolation of individual clones allowed for insert size profiling at Day 7. Screened isolates from Day 28 did not reveal any galactokinase inserts.

C   Functional characterization in minimal media with galactose as the sole carbon source. Three sets of strains were studied: (i) starting *E. coli* strains transformed with the CDS of each gene cloned into the backbone vector, (ii) *E. coli* clones directly isolated from stool samples, and (iii) starting *E. coli* strains re-transformed with individual plasmids isolated from stool samples. All clones isolated at Day 28 carried the BT_1759 locus. Lines represent the mean.

D   Genotyping of the background *E. coli* genome at the *galK* locus. 20 clones isolated from mice inoculated with the library at each of the indicated time points were screened, while 30 clones isolated from the lux control mice at Day 28 were screened.

Source data are available online for this figure.

all of our isolates were Lac⁻. However, the *lacY* (F27S) mutant reached a higher density in M9 galactose compared to other Day 28 isolates, which also carried the same plasmid-borne Bt glycoside hydrolase (Supplementary Fig S5A). This clone also grew to a greater density than *E. coli* recipient strains in which we had cloned the Bt galactokinase operon (BT_0370-BT_0372) (Supplementary Fig S5B). The *lacY* transporter can transport galactose in addition to lactose, and *lacY* mutants have been shown previously to confer faster growth of *E. coli* MG1655 on galactose (Soupene *et al*, 2003).

Remarkably, we found an interaction between the *E. coli* genome and a heterologously expressed Bt gene. Isolate 3 from Mouse 1 on Day 7 had an SNV in the galactose repressor, *galR* (R20L), in its DNA-binding domain (Weickert & Adhyat, 1992). *Escherichia coli* GalR binds operator sequences upstream of the *galETK* operon (Weickert & Adhya, 1993), and the amino acid substitution of arginine for leucine could be disruptive to binding. Using MacConkey galactose plates, we found that the *galR* (R20L) isolate was Gal⁺, whereas a similar Day 7 clone, which also had a genomic *galK*⁻ genotype and a Bt galactokinase (BT_0370) insert but no *galR* SNV, exhibited

**Table 2. Genetic variants in mouse-isolated clones identified by whole-genome sequencing.**

| Sample | Insert locus and size (kb) | Genomic galK+/− | Variant position on *Escherichia coli* genome | Variant impact and coverage |
|---|---|---|---|---|
| NEB Turbo control | − | galK− | | |
| Day 7 Mouse 1 clone 1 | BT_1759 (2.5) | galK− | | |
| Day 7 Mouse 1 clone 3 | BT_0370 (4.0) | galK− | SNV 2976657 G>T | galR (R20L) [34/34 reads] |
| Day 7 Mouse 2 clone 5 | BT_0370 (4.3) | galK− | | |
| Day 7 Mouse 3 clone 1 | BT_1759 (3.1) | galK− | | |
| Day 7 Mouse 4 clone 4 | BT_0370 (4.1) | galK− | | |
| Day 7 Mouse 5 clone 2 | BT_1759 (2.5) | galK+ | | |
| Day 7 Mouse 5 clone 4 | BT_1759 (3.1) | galK− | | |
| Day 28 Mouse 1 clone 1 | BT_1759 (2.5) | galK+ | SNV 363100 A>G | lacY (F27S) [173/173 reads] |
| Day 28 Mouse 2 clone 1 | BT_1759 (2.5) | galK+ | | |
| Day 28 Mouse 3 clone 1 | BT_1759 (2.5) | galK+ | | |
| Day 28 Mouse 4 clone 1 | BT_1759 (2.5) | galK+ | | |
| Day 28 Mouse 5 clone 1 | BT_1759 (2.5) | galK+ | | |
| Day 28 Mouse 5 clone 4 | BT_1759 (4.2) | galK+ | | |
| Day 28 Mouse 7 clone 1 lux control | − | galK+ | | |
| Day 28 Mouse 10 clone 2 lux control | − | galK− | 1  SNV 3991675 G>A<br>2  SNV 780994 G>A<br>3  F plasmid SNV 67772 T>C | 1  cyaA (G175S) [122/122 reads]<br>2  intergenic, between lysW and valZ [108/108 reads]<br>3  traY promoter (−35) [229/229 reads] |

a Gal⁻ phenotype. Since the BT_0370 inserts in the Day 7 clones were not identical (Fig 5B), we re-transformed the plasmids into the starting *E. coli* strain to confirm the phenotype and rule out effects from an underlying chromosomal *galR* mutation. In M9 galactose medium, the *galR* (R20L) mutant grew to a higher cell density than a wild-type *galR* strain with the same Bt galactokinase plasmid (Supplementary Fig S5C). These findings indicate that the *E. coli* genome had co-evolved with the *in vivo* selection of plasmids carrying Bt genes for galactose utilization.

We found no mutations in the library plasmids or Bt genes, and, aside from loss of the IS2 element in the *galK* gene, all other IS elements were intact on the *E. coli* genome. In aggregate, these small numbers of variants ($\sim$10$^{-8}$ mutations per bp per day) in the *E. coli* recipient strain suggest that outside of genetic loci with selective pressures exerted upon them, the organism remained genetically stable in the mammalian gut over the course of our experiment.

## Discussion

We have demonstrated the use of TFUMseq for high-throughput *in vivo* screening of genetic fragments from an entire donor genome from a commensal microbe to increase the fitness of a phylogenetically distant bacterial species in the mammalian gut. To our knowledge, this is the first demonstration of temporal functional metagenomics using shotgun libraries applied to the *in vivo* mammalian gut environment. Our findings attest to the value of a time-series approach, as the shifts in population dynamics of clones harboring different gene fragments would not have been discovered if we had only obtained endpoint data. Further, we introduced computational methods using information theoretic measures and statistical longitudinal analysis techniques that allowed us to identify and localize significant selection of donor genes over time.

In this demonstration of the TFUMseq approach using an *E. coli* plasmid library of Bt genes, we uncovered sequential selection of clones with different carbohydrate utilization genes—first for galactose and then for sucrose metabolism. Galactose plays a substantial role in selection in our experiment, as all three of the observed *E. coli* genomic mutations (in *galK*, *lacY*, and *galR*) affected galactose utilization, and we observed selection for Bt galactokinase (BT_0370) and glucose/galactose transporter (BT_0371) *in vivo*. Galactose is a component of the hemi-cellulose that makes up part of the 15.2% neutral detergent fiber in mouse chow, although galactose composition was not explicitly provided by the manufacturer. Galactose is also a component of mammalian mucin in the GI tract (Juge, 2012). However, our observation that *in vitro* selection occurs for the BT_0370 and BT_0371 galactose utilization locus in MC medium indicates that the mouse chow diet itself is providing sufficient galactose to exert selective pressure. During our *in vivo* colonization experiments, once *E. coli* restored native galactokinase (*galK*)

activity in its genome through loss of IS2, Bt genes that catabolized a second carbon source, sucrose, became dominant. Sucrose is a major simple carbohydrate in mouse chow, present at 0.71% (w/w) in comparison with 0.22% for glucose and fructose. Per Freter's nutrient-niche hypothesis, which described substrate-level competition and substrate-limited population levels (Freter *et al*, 1983), our results suggest that galactose is preferred over sucrose and that a clone capable of utilizing both carbon sources will outcompete clones capable of only using one of the sources. Nutrient-based niches have been documented in the mammalian GI tract, including the varying sugar preferences among commensal and pathogenic *E. coli* strains (Maltby *et al*, 2013), and polysaccharide utilization loci (PULs) in *Bacteroides* species that promote long-term colonization (Lee *et al*, 2013). In fact, the enterohemorrhagic *E. coli* strain EDL933 can use sucrose, while commensal *E. coli* strains K-12 MG1655, HS, and Nissle 1917 cannot (Maltby *et al*, 2013). Incorporating sucrose utilization, such as through the truncated Bt glycoside hydrolase (BT_1759) identified in this study, could enhance retention of probiotic *E. coli* strains. Pre-colonization with sucrose-utilizing probiotic strains to occupy the sucrose niche could also be an effective strategy to resist pathogen colonization.

*Bacteroides thetaiotaomicron* has been investigated previously using transposon mutagenesis systems coupled to mouse gut colonization experiments (Goodman *et al*, 2009), facilitating comparison of our results to the prior study. Goodman *et al* found no difference in the abundances of galactokinase (BT_0370) mutants *in vitro*, but BT_0370 mutants were underrepresented *in vivo*. In contrast, in our study, the Bt galactokinase was selected not only *in vivo*, but also *in vitro*. Furthermore, Goodman *et al* found dTDP-4-dehydrorhamnose reductase (BT_1730) and GMP synthase (BT_4265) mutants were underrepresented both *in vitro* and *in vivo*. However, in our study, BT_1730 and BT_4265 seemed to confer fitness only *in vivo*. The *in vitro* discrepancies may be a result of slightly different culturing and media conditions. The *in vivo* results are in agreement for BT_0370, BT_1730, and BT_4265, though the other genes we identified in our experiments were not significantly altered in representation in the transposon mutagenesis experiments, highlighting the different capabilities of the two approaches.

Overall, we expect TFUMseq to be a powerful tool for engineering commensal microbes with new or enhanced capabilities, as it provides a general approach to functionally identifying genes from metagenomic DNA that enhance microbial fitness *in vivo*. Going forward, there are two primary considerations for designing future TFUMseq experiments: the choice of the bacterial strain to receive the donor plasmid library and the mammalian host environment. In this study, we used a cloning strain of *E. coli* as the recipient bacteria, which enabled the generation of a robust, high-quality library. This strain has inactivated restriction systems, thus preventing underrepresentation of DNA inserts in the library that may contain otherwise recognized methylated sites from the donor source. Further, the lack of prior host adaptation of this laboratory strain *in vivo*, in comparison with a wild-type adapted commensal strain, allows for stronger selection signals from clones harboring functional donor genes. As we saw, the recipient strain also plays a role in the co-evolution of the insert library and the bacterial genome. We observed a genomic change, specifically the *galK* reversion, driving the shift in

library selection from Bt galactokinase (BT_0370) to Bt glycoside hydrolase (BT_1759). Furthermore, we found single-nucleotide variations in *E. coli galR* and *lacY* loci that boosted galactose utilization in clones harboring functional Bt genes. Given that co-evolution drives genomic changes in the recipient strain, using a well-characterized recipient strain facilitates mechanistic interpretation of these changes.

The state of the mammalian host is also a critical variable in our approach. In this work, germfree mice were mono-associated with the library. We expect that the results of *in vivo* selection may differ when mice are pre-colonized with a microbiota due to changes in nutrient availability and other ecological interactions, including competition or syntrophy. For instance, co-colonization experiments demonstrated that probiotic strains and commensal bacteria have adaptive substrate utilization. Bt shifts its metabolism from mucosal glycans to dietary plant polysaccharides when in the presence of *Bifidobacterium animalis*, *Bifidobacterium longum*, or *Lactobacillus casei* (Sonnenburg *et al*, 2006). *Bacteroides* species are also known to engage in public goods-based syntrophy by releasing outermembrane vesicles (OMVs) that contain surface glycoside hydrolases or polysaccharide lyases (Rakoff-Nahoum *et al*, 2014). These enzymes catabolize large polysaccharides into smaller units, which can then be utilized by other species in the community. Given the complexities of multispecies bacterial communities, TFUMseq's ability to track large numbers of clones over time will be important for detecting relevant genes that confer a fitness advantage within dynamically changing communities.

Our results suggest several future studies using TFUMseq. Replication of our experiments in additional cohorts of mice would be valuable. In this study, mice were separately caged in the same gnotobiotic isolator, and we employed meticulous techniques to avoid cross-contamination. We did not observe evidence of isolates being exchanged between mice and, in fact, saw unique selection patterns for each mouse (Supplementary Fig S3) and were able to isolate different clones carrying non-identical fragments from different mice. Nonetheless, future experiments in which our study is repeated in different gnotobiotic isolators would be useful to characterize the variability of the entire process. Further, it would be of interest to understand the influence of host genetics and nutrition on the selection of genes in our library, which could be investigated by repeating our study using different strains of mice or placing mice on different diets such as high-fat/high-sugar chow. Also, potential investigations could use total metagenomic DNA from stool samples, rather than DNA from cultured organisms. Another area of interest would be probing community composition and dynamics of selection in different regions of the gut. These studies would provide insights into biogeographical niches coupled with temporal data provided by our method. TFUMseq could also be used to build a better probiotic strain. One could incorporate a metagenomic plasmid library into a probiotic strain and introduce the strain into a complex host–bacterial community to isolate genes that increase the strain's fitness *in vivo*. We have already identified sucrose utilization as an important and feasible trait to incorporate into an enhanced probiotic strain. Ultimately, TFUMseq-based studies could enable the rational design of probiotic or commensal strains for various clinical applications, such as resisting pathogen colonization, compensating for a high-fat/high-sucrose Western diet, or tempering host autoimmunity.

## Materials and Methods

### Bacterial strains and growth conditions

*Bacteroides thetaiotaomicron* VPI-5482 (ATCC # 29148) was grown anaerobically in a rich medium based on brain heart infusion with other supplements added (see Supplementary Materials and Methods). The genomic library was maintained in an *Escherichia coli* K-12 strain, NEB Turbo (New England Biolabs, Ipswich, MA, USA). *Escherichia coli* strains were grown in Luria broth (LB) and supplemented with carbenicillin (final concentration 100 μg/ml) as needed. For anaerobic growth, an anaerobic jar (GasPak System, Becton Dickinson, Franklin Lakes, NJ) was used. Mouse chow (MC) filtrate was prepared by adding 150 ml deionized water to 8 g of crushed mouse chow (Mouse Breeding Diet 5021, LabDiet, St. Louis, MO, USA). The mixture was heated at 95°C for 30 min with mixing, passed through a 0.22-μm filter, and autoclaved. The sterility of the MC filtrate was confirmed by incubating at 37°C in aerobic and anaerobic conditions and observing no growth after several days.

### Library generation

*Bacteroides thetaiotaomicron* genomic DNA was isolated (DNeasy Blood & Tissue kit, Qiagen, Venlo, The Netherlands), fragmented by sonication to 3–5 kb (Covaris E210, Covaris, Woburn, MA, USA), and size-selected and extracted by gel electrophoresis (Pippin Prep, Sage Sciences). The fragments were end-repaired (End-It DNA End-Repair kit, Epicenter, Madison, WI, USA) and cloned into a PCR-amplified GMV1c backbone vector via blunt-end ligation. The reaction was transformed into NEB Turbo electrocompetent *E. coli* cells (New England Biolabs). The library size was quantified by counting colonies formed on selective media (LB carbenicillin) after plating a fraction of the transformed cells. To assess the size of inserts successfully cloned into the library, we picked colonies for PCR amplification using primers ver2_f/r (Supplementary Table S2) that flanked the insert site. We further confirmed the presence of inserts by submitting amplified inserts for Sanger sequencing (Genewiz, South Plainfield, NJ, USA) and aligning sequences with the donor *B. thetaiotaomicron* genome. A more detailed protocol is included in the Supplementary Materials and Methods.

### Plasmid retention

Individual stool pellets from Days 0.75, 1.5, 1.75, 2.5, 4, 10, 14, 21, 25, and 28 were homogenized in 10% PBS and plated on LB agar with or without carbenicillin (carb). To obtain accurate counts, colony platings were performed in triplicate and repeated at 100× dilutions if the plates were overgrown. Plasmid retention was calculated as the number of colonies grown on LB-carb plates divided by the number of colonies grown on LB only plates.

### *In vitro* selection

After inoculating the library in LB or MC broth, the cultures were passaged by diluting at 20× into fresh media. LB cultures were grown in aerobic conditions with shaking and passaged every day for 2 weeks. MC cultures were grown in anaerobic conditions without shaking and passaged every 2 days for 2 weeks, since the cultures took more time to reach saturation compared to the LB condition.

### *In vivo* selection

All of the mice used in this study were handled in accordance with protocols approved by the Harvard Medical Area Standing Committee on Animals (HMA IACUC). Male C57BL/6 mice, 6–8 weeks of age, were used. The mice were bred in the Center for Clinical and Translational Metagenomics facility and maintained in germfree conditions prior to the experiments. Germfree mice were orally gavaged with ~2 × $10^8$ CFU of bacteria in a volume of 200 μl on Day 0. Mice receiving the library were individually caged in a gnotobiotic isolator. Control mice were maintained in a single cage in a separate gnotobiotic isolator. Fecal pellets were collected at 0.5, 0.75, 1.5, 1.75, 2.5, 3, 4, 7, 10, 14, 21, 25, and 28 days post-inoculation and stored at −80°C in 10% PBS buffer.

### Colony PCR and Sanger sequencing

Individual colonies were isolated from stool samples streaked onto LB agar with carbenicillin (100 μg/ml). Colonies were grown overnight at 37°C in a 96-well plate with 200 μl of LB + carbenicillin. 0.8 μl of the culture was added to a total PCR volume of 20 μl. The PCR mix (KAPA HiFi HotStart ReadyMix PCR kit, Kapa Biosystems, Wilmington, MA, USA) contained primers ver2_f/r (Supplementary Table S2) that flanked the insert site. PCR amplicons were submitted for sequencing (Genewiz), and the insert sequence was mapped back to the *B. thetaiotaomicron* genome using BLASTn. Primers for genotyping the *galK* locus on the *E. coli* genome are listed in Supplementary Table S2. The presence or absence of IS2 in *galK* was confirmed using primers galK16_chk_f/r that flanked the expected insertion site in *galK*.

### DNA extraction and PCR amplification of inserts for Illumina sequencing

DNA was extracted from collected samples in the *in vitro* experiment using the DNeasy Blood & Tissue kit (Qiagen). Inserts were PCR-amplified using primers ver2_f/r (Supplementary Table S2) in KAPA HiFi HotStart Mix (Kapa Biosystems) and purified with Agencourt AMPure XP beads (Beckman Coulter, Indianapolis, IN, USA) at a beads:sample volumetric ratio of 0.5:1. The amplicons were prepared for sequencing using the Nextera kit (Illumina, San Diego, CA, USA). For all fecal samples from the *in vivo* experiment, the QIAamp DNA Stool Mini kit (Qiagen) was used. Isolated DNA was digested with PspXI and AvrII enzymes (New England Biolabs) prior to purification with QIAquick PCR Purification kit (Qiagen) and subsequent PCR amplification with primers A_L and A_R (Supplementary Table S2) in KAPA HiFi HotStart Mix. The PCR was purified with AMPure beads at a beads:sample volumetric ratio of 0:5:1. A detailed protocol is included in the Supplementary Materials and Methods.

Initially, in our sequencing of the *in vitro* samples, we observed a high fraction (30–45%) of reads mapping to the backbone vector and fewer reads (20%) mapping to the *B. thetaiotaomicron* genome. Given the large (> 3 kb) insert sizes of these libraries, traditional amplification methods evidently over-amplify the smaller vector backbone (2 kb), thereby overwhelming vectors containing actual

genomic inserts. We therefore optimized the sample preparation protocol by incorporating a double digestion strategy prior to PCR amplification of the inserts (Supplementary Fig S6). The two restriction sequences selected were the least common (of all available sites on the plasmid) in the *B. thetaiotaomicron* genome. With this new protocol, in our subsequent *in vivo* sequencing, we observed < 4% of reads mapping to the backbone vector and > 90% of reads mapping to the *B. thetaiotaomicron* genome.

### High-throughput sequencing and analysis of *in vitro* library selection data

Samples were sequenced on the MiSeq (Illumina) instrument at the Molecular Biology Core Facilities of the Dana-Farber Cancer Institute. Metrics for this sequencing run are provided in Supplementary Table S3. Due to the PCR amplification protocol prior to optimization (see previous section), we observed large amounts of *E. coli* plasmid DNA in our sequencing reads. To maximize the reads aligned to the *B. thetaiotaomicron* genome, we aggressively trimmed low-quality bases and removed sequences mapping to the *E. coli* genome or with length shorter than 20 nt. The reference genome of *B. thetaiotaomicron* (NC_004663 and NC_004703) was downloaded from the NCBI nucleotide database (http://www.ncbi.nlm.nih.gov/). Due to the aggressive preprocessing of reads described, the length of trimmed sequences was shorter than 50 nt. Therefore, Bowtie (Langmead *et al*, 2009) was applied instead of Bowtie2 for higher sensitivity. Default parameters were used for building a Bowtie index with the *B. thetaiotaomicron* chromosome and plasmid sequences. Paired-end reads were aligned to the reference genome with parameter –X 300 using Bowtie. SAM files from the Bowtie alignment were converted to indexed and sorted BAM files using SAMtools (Li *et al*, 2009). Cuffdiff (Trapnell *et al*, 2013) was applied to test for differential representation of genes (i.e. the library grown in rich medium at time 0 versus the library grown in rich medium at Day 7 and the library grown in MC medium at time 0 versus the library grown in MC medium at Day 6).

### High-throughput sequencing and processing of *in vivo* library selection data

*Bacteroides thetaiotaomicron* genomic DNA inserts were amplified from isolated *E. coli* plasmids using our improved PCR protocol (Supplementary Fig S6). After Nextera sequencing library preparation, paired-end reads of 101 nt length were generated on the HiSeq 2500 (Illumina) instrument at the Baylor College of Medicine Alkek Center for Metagenomics and Microbiome Research. Metrics for this sequencing run are provided in Supplementary Table S4. All reads passed quality control (base quality > 30) using FastQC (http://www.bioinformatics.babraham.ac.uk/projects/fastqc/). To eliminate plasmid DNA sequences in reads, the reads were trimmed using custom Perl scripts that removed all flanking regions matching 15 bp of the plasmid DNA on the 5′ and 3′ ends of insert fragment. Reads < 20 bp after trimming were discarded, and the others were matched as pairs with the forward read and reverse reads.

Sequencing reads were mapped onto the reference genome of *B. thetaiotaomicron* using Bowtie2 (Langmead *et al*, 2009). Default parameters were used for building the Bowtie2 index using the *B. thetaiotaomicron* chromosome and plasmid sequences, and for

aligning reads to the reference sequence. SAM files generated from Bowtie2 alignment were converted to indexed and sorted BAM files using SAMtools (Li *et al*, 2009). In SAMtools, "mpileup" with parameter "-B" was used to obtain the depth of coverage of the reference genome. Across all samples, the mean of the mapped bases to the *B. thetaiotaomicron* genome was $1.17 \times 10^9$, with a minimum of $4.31 \times 10^8$ and maximum of $2.49 \times 10^9$ bases per sample.

### Statistical analyses of *in vivo* selection data

Analyses were performed using custom functions written in MATLAB (MathWorks, Natick, MA, USA). The effective positional diversity (EPD), a genome-wide measure of the diversity of library representation, was calculated using the formula:

$$\text{EPD}(t) = e^{-\sum_{i=1}^{P} r_{ti} \ln r_{ti}}$$

Here, $r_{ti}$ represents the fraction of reads at time $t$ mapping to nucleotide $i$ in a reference sequence totaling $P$ nucleotides (e.g. the Bt genome).

The time-averaged relative abundance (TA-RA), a gene-level measure of library selection, was calculated using the formula:

$$\text{TA-RA}(g, t_1, t_2) = \int_{t_1}^{t_2} \frac{f_g(\tau) d\tau}{t_2 - t_1}$$

Here, $t_1$ and $t_2$ denote the bounds of the time interval of interest, and $f_g$ represents a continuous-time function for gene $g$. The function $f_g$ was estimated as follows. We fit a cubic smoothing spline, using the MATLAB function `csaps`, applied to the log fold change in fragments per kilobase per million (FPKM) mapped reads for gene $g$ at each time point $t$ (i.e. the FPKM value at time point $t$ divided by the FPKM value for the gene in the starting library). FPKM values were generated using Cufflinks (Trapnell *et al*, 2013) with parameter –max-bundle-frags 40,000,000. The smoothing spline was used to account for non-uniform temporal sampling and noise in the data.

The time-averaged normalized effective coverage (TA-NEC), a gene-level measure of coverage, was calculated using the formula:

$$\text{TA-NEC}(g, t_1, t_2) = \int_{t_1}^{t_2} \frac{h_g(\tau) d\tau}{(t_2 - t_1) l_g}$$

Here, $l_g$ denotes the length of gene $g$, and $h_g$ represents a continuous-time function for gene $g$. The function $h_g$ was estimated as follows. We fit a cubic smoothing spline, using the MATLAB function `csaps`, applied to the effective coverage, EC($g,t$), for the gene at each time point:

$$\text{EC}(g, t) = e^{-\sum_{i=s_g}^{s_g+l_g} r_{ti} \ln r_{ti}}$$

Here, $s_g$ denotes the start of the gene.

To detect genes with significantly higher than expected selection, we performed a one-sided *t*-test on Box-Cox-transformed TA-RA and TA-NEC values and corrected for multiple hypothesis testing using the MATLAB function `mafdr`. To estimate the relevant null

hypotheses for the *t*-tests, while taking into account possible biases due to differential representation of genes in the input library, we used a robust regression algorithm (MATLAB function `robustfit`) in which the input library value served as the independent variable and the TA-RA or TA-NEC value served as the dependent variable.

### Whole-genome sequencing of isolated clones from the *in vivo* library selection

Whole-genome sequencing of *E. coli* recipient isolates from seven Day 7 clones, six Day 28 clones, and two Day 28 luciferase control clones was performed on the MiSeq (Illumina) instrument after Nextera (Illumina) sequencing library preparation at the Molecular Biology Core Facilities of the Dana-Farber Cancer Institute. Metrics for this sequencing run are provided in Supplementary Table S5. The raw data were processed with Millstone (http://churchlab.github.io/millstone), which combines BWA alignment, GATK for BAM realignment and cleanup, and SnpEff for variant effect prediction. Reads were aligned to *E. coli* K-12 DH10B as well as MG1655 to identify any variants not in common with the starting library strain NEB Turbo. The average genome coverage of each sequenced strain ranged from 20 to 140×. Alignments were also performed against the F plasmid (which is present in the starting recipient strain) and a library plasmid with the expected insert (which we had characterized in Sanger sequencing of individual clones).

### Growth assays

Cells were pre-conditioned by growth in minimal media (M9) supplemented with 0.2% glucose prior to inoculation in growth assays. Then, 1 μl of the culture was inoculated into a final volume of 200 μl of M9 supplemented with 0.2%, unless otherwise noted, of a sole carbon source, such as glucose, lactose, galactose, or sucrose. When needed, MacConkey base agar with a final concentration of 1% lactose or galactose was also used to characterize lactose or galactose utilization.

### Data availability

All sequencing data generated in this study are publicly available at NCBI SRA under accession number SRP051326. Calculated effective gene coverage and FPKM values for each gene and mouse from the *in vivo* experiment are available as Supplementary Datasets S1 and S2.

**Supplementary information** for this article is available online: http://msb.embopress.org

### Acknowledgements
This work was supported by grants from the Harvard Digestive Diseases Center (Pilot and Feasibility Grant to GKG, under NIH Grant P30DK034854), the National Institutes of Health Director's Early Independence Award (Grant 1DP5OD009172-01 to HHW), the US Department of Energy (Grant DE-FG02-02ER63445 to GMC), and the Wyss Institute for Biologically Inspired Engineering. SJY also acknowledges support from the National Science Foundation Graduate Research Fellowship and the MIT Neurometrix Presidential Graduate Fellowship. GKG also acknowledges support from the Brigham and Women's Department of Pathology.

### Author contributions
SJY, LD, JLB, LB, HHW and GKG designed the experiments. SJY, LD, JLB, LB and HHW performed the experiments. SJY, LD, NL, JLB, GMC, LB, HHW and GKG analyzed the data. SJY, NL, JLB, LB, HHW and GKG contributed to writing of the manuscript.

### Conflict of interest
The authors declare that they have no conflict of interest.

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
