## [Review Process File · Molecular Systems Biology]

Improving microbial fitness in the mammalian gut by in vivo temporal functional metagenomics

Stephanie J. Yaung, Luxue Deng, Ning Li, Jonathan L. Braff, George M. Church, Lynn Bry, Harris H. Wang and Georg K Gerber

Corresponding author: Harris Wang, Columbia University

Review timeline:	Submission date:	23 October 2014
	Editorial Decision:	12 December 2014
	Revision received:	08 January 2015
	Editorial Decision:	19 January 2015
	Revision received:	21 January 2015
	Accepted:	23 January 2015

Editor: Maria Polychronidou

Transaction Report:

1st Editorial Decision

12 December 2014

Thank you again for submitting your work to Molecular Systems Biology. First of all, I would like to apologize for the somewhat delayed response. We have now heard back from two of the three referees who agreed to evaluate your manuscript. Unfortunately, after several reminders we have still not received a report from reviewer #2. Since the recommendations of the other two referees are similar, I prefer to make a decision now rather than further delaying the process.

Overall, the reviewers think that the presented approach seems interesting. However, they list a series of issues that we would ask you to address in a revision of this work. Their recommendations are clear so there is no need to repeat the points listed below. On a more minor note, related to comment #2 of reviewer #1, we think that using the name TFUMseq to describe the method is not an issue.

On a more editorial level, we would like to ask you to include a "Data Availability" section, providing the accession numbers and all other relevant information related to the newly generated sequencing data.

REFeree REPORTS

Reviewer #1:

"Improving microbial fitness in the mammalian gut by in vivo temporal functional metagenomics" aims to define bacterial gene function by cloning fragments of bacterial genomes into *E. coli* and assaying the enrichment/depletion (i.e., the selection/fitness) of each insert in the context of a diverse community of inserts. Several laboratories have done this type of work in vitro, particularly in the case of antibiotic resistance, while in vivo (i.e., in a murine host) studies have to date focused on determining gene function in a high-throughput manner by depleting genes in the strain of interest. This manuscript thus provides the first glimpse of the potential of such an approach for better understanding microbiome functional. The method has great advantages of simplicity (e.g., unlike the knockout type assays, as in Goodman et.al., 2009, you do not have to develop new genetic tools) with the disadvantage of limited insert lengths (could be overcome) and a non-native host expressing the inserts. Overall, I think the simplicity of this approach and the flexibility to include gene fragments from multiple microbes and whole communities will make this a technique that other labs employ as well. I have only a few critiques.

1) There is no mention of how these large fragments are run on an Illumina sequencer. Were any modifications to the sequencing protocols performed to enable clustering of such large fragments? In my own experience of trying to sequence large fragments (2-3 years ago), it was difficult to get clusters on the Illumina flow cell much larger than 1kb. Perhaps the technology is more forgiving now or my methods were flawed, but an explanation would be useful in the text somewhere.

2) I'm not sure we need to give a name of TFUMseq to this method. Cloning genes to look for function is a pretty old idea. The novelty is how the method is employed.

3) It is unfortunate the authors didn't do a complete repeat of the experiment in a second batch of gnotobiotic animals. While individual housing will provide some independence between the animals, it is still hard to gauge the repeatability of these experiments unless the entire process is repeated.

4) I really appreciate the well-written methods in the expanded view. If all manuscripts were so forthright and comprehensive with the details of their techniques, scientists would benefit greatly.

Reviewer #3:

This manuscript describes an in vivo functional metagenomics approach that enables the identification of genomic fragments conferring a selective advantage to a carrier bacterium in its host (in this case, mouse). The technique is elegant and as strong potential as a method to detect specific genes associated with colonization success under a wide range of imaginable circumstances. The paper is interesting, well written and I didn't spot any major flaws. The only issues worth mentioning are:

- the introduction discusses the problems with functional assessment of unknown genes at length. However, the approach presented by the authors does not solve this (important) issue directly - it merely helps identifying genes conferring a fitness advantage in a specific context and will thus only serve for functional annotation of specific genes in specific cases.
- Also, the intro largely ignores the bulk of pioneering functional metagenomics studies (of which this is a derivative adapted to host-associated communities), by a.o the Handelsman lab. This should be corrected.

1st Revision - authors' response

08 January 2015

Reviewer #1:

"Improving microbial fitness in the mammalian gut by in vivo temporal functional metagenomics" aims to define bacterial gene function by cloning fragments of bacterial genomes into E. coli and assaying the enrichment/depletion (i.e., the selection/fitness) of each insert in the context of a diverse community of inserts. Several laboratories have done this type of work in vitro, particularly in the case of antibiotic resistance, while in vivo (i.e., in a murine host) studies have to date focused on determining gene function in a high-throughput manner by depleting genes in the strain of interest. This manuscript thus provides the first glimpse of the potential of such an approach for

better understanding microbiome functional. The method has great advantages of simplicity (e.g., unlike the knockout type assays, as in Goodman et al., 2009, you do not have to develop new genetic tools) with the disadvantage of limited insert lengths (could be overcome) and a non-native host expressing the inserts. Overall, I think the simplicity of this approach and the flexibility to include gene fragments from multiple microbes and whole communities will make this a technique that other labs employ as well. I have only a few critiques.

1) There is no mention of how these large fragments are run on an Illumina sequencer. Were any modifications to the sequencing protocols performed to enable clustering of such large fragments? In my own experience of trying to sequence large fragments (2-3 years ago), it was difficult to get clusters on the Illumina flow cell much larger than 1kb. Perhaps the technology is more forgiving now or my methods were flawed, but an explanation would be useful in the text somewhere.

We apologize for any confusion regarding our sequencing strategy. We did not modify the sequencing protocol to enable clustering of large fragments. Instead, we selectively amplified the large inserts using PCR and submitted the amplicons for standard Illumina sample prep (with the Nextera tagmentation kit) and sequencing. For the sequencing reads, we trimmed off any sequences from the backbone vector and mapped the remaining sequence to the *B. thetaiotaomicron* genome. Our computational approach allows us to detect selected genes in the donor genome that are uniformly covered over time by more than the expected background number of sequencing reads. Further, we performed sequencing with sufficient depth to resolve the structure of large fragments containing multiple genes (i.e., map reads to the junctions between genes) in dominant clones in the selection.

We have added an additional panel to Figure E6 and text in the results section (lines 188-194) to clarify our approach: “To characterize the entire *in vivo* selected library over time, we extracted DNA from collected stool samples, PCR amplified the donor inserts, prepared sequencing libraries of the amplicons, sequenced libraries on the Illumina HiSeq 2500 instrument, and used computational techniques to detect selected genes in the donor genome that were uniformly covered over time by more than the expected background number of sequencing reads. Each sample resulted in ~7 million 101 nt paired-end reads (Table E4) that were mapped back to the donor genome (Figure E6).”

2) I'm not sure we need to give a name of TFUMseq to this method. Cloning genes to look for function is a pretty old idea. The novelty is how the method is employed.

We appreciate this comment, and agree that the novelty of our approach involves leveraging well-established techniques for new applications. However, we believe that for future work, the name TFUMseq will be useful for describing the particular combination of techniques that we have introduced, involving temporal tracking of clones and a high-throughput sequencing and computational approach to identify selected genes.

3) It is unfortunate the authors didn't do a complete repeat of the experiment in a second batch of gnotobiotic animals. While individual housing will provide some independence between the animals, it is still hard to gauge the repeatability of these experiments unless the entire process is repeated.

We appreciate the reviewer's comments and agree that numerous factors can affect the reproducibility of gnotobiotic experiments. At the Center for Clinical and Translational Metagenomics, we have extensive experience with these issues, and have successfully employed separate caging for biological replicates in numerous other experiments. We use meticulous techniques to avoid cross-contamination among the cages, including using separate sterile materials each time mice are handled or stool collections are performed. Additionally, from our detailed temporal tracking of clones in samples from each mouse, we did not observe evidence of isolates being exchanged between mice. In fact, we observed unique selection patterns for each mouse (Figure E3), and were able to isolate different clones carrying non-identical fragments from different mice. For these reasons, we are confident that we achieved sufficient independence among the mice to constitute biological replicates. Nonetheless, the reviewer raises an important issue, and we agree that additional gnotobiotic experiments to verify our results, perhaps extending to different mouse strains or diets, is an important direction for future work.

We have incorporated the reviewer's comments and suggestions in the discussion section (lines 489-499): "Replication of our experiments in additional cohorts of mice would be valuable. In this study, mice were separately caged in the same gnotobiotic isolator, and we employed meticulous techniques to avoid cross-contamination. We did not observe evidence of isolates being exchanged between mice, and in fact, saw unique selection patterns for each mouse (Figure E3) and were able to isolate different clones carrying non-identical fragments from different mice. Nonetheless, future experiments in which our study was repeated in a different gnotobiotic isolator would be useful to characterize the variability of the entire process. Further, it would be of interest to understand the influence of host genetics and nutrition on the selection of genes in our library, which could be investigated by repeating our study using different strains of mice or placing mice on different diets such as high-fat/high-sugar chow."

4) I really appreciate the well-written methods in the expanded view. If all manuscripts were so forthright and comprehensive with the details of their techniques, scientists would benefit greatly.

We are grateful for the reviewer's remark, and hope that our detailed explication of our methods will benefit other research groups.

Reviewer #3:

This manuscript describes an in vivo functional metagenomics approach that enables the identification of genomic fragments conferring a selective advantage to a carrier bacterium in its host (in this case, mouse). The technique is elegant and as strong potential as a method to detect specific genes associated with colonization success under a wide range of imaginable circumstances. The paper is interesting, well written and I didn't spot any major flaws. The only issues worth mentioning are: - the introduction discusses the problems with functional assessment of unknown genes at length. However, the approach presented by the authors does not solve this (important) issue directly - it merely helps identifying genes conferring a fitness advantage in a specific context and will thus only serve for functional annotation of specific genes in specific cases.

We appreciate the reviewer's critique. We agree that our method enriches for specific genes in specific contexts, which is the purpose of our functional metagenomics approach, and that it is meant as a first-pass screen for further follow up studies, which we performed for several of the putatively selected genes in this work.

Based on the reviewer's comments, we have changed the introduction to emphasize these points (lines 89-92): "Our approach uses physical shearing or restriction digestion of donor DNA to generate fragments that are cloned into an expression vector and transformed into the recipient bacterial strain, for high-throughput functional screening to identify genes that confer a fitness advantage in a particular context."

- Also, the intro largely ignores the bulk of pioneering functional metagenomics studies (of which this is a derivative adapted to host-associated communities), by a.o the Handelsman lab. This should be corrected.

We thank the reviewer for pointing out our omission in the introduction section of the pioneering studies by the Handelsman lab, and have now incorporated references to this work into the introduction section (lines 95-98): "Functional metagenomics using environmental samples was first established for communities derived from lignocellulosic feedstocks (Healy *et al*, 1995), seawater (Stein *et al*, 1996), and soil (Rondon *et al*, 2000)."

Thank you again for submitting your work to Molecular Systems Biology. We have now heard back from the referee who was asked to evaluate your manuscript. As you will see below, this referee is satisfied with the modifications made and thinks that the study is now suitable for publication.

REFEREE REPORTS

Reviewer #1:

The study is now suitable for publication.